# The Rise of AI Language Pathologists: Exploring Two-level Prompt Learning for Few-shot Weakly-supervised Whole Slide Image Classification

**Linhao Qu**[1,2]**, Xiaoyuan Luo**[1,2]**, Kexue Fu**[1,2]**, Manning Wang**[*1,2]**, Zhijian Song**[*1,2]
[1]Digital Medical Research Center, School of Basic Medical Science, Fudan University.
[2] Shanghai Key Lab of Medical Image Computing and Computer Assisted Intervention.

## Abstract

This paper introduces the novel concept of few-shot weakly supervised learning for pathology Whole Slide Image (WSI) classification, denoted as FSWC. A solution is proposed based on prompt learning and the utilization of a large language model, GPT-4. Since a WSI is too large and needs to be divided into patches for processing, WSI classification is commonly approached as a Multiple Instance Learning (MIL) problem. In this context, each WSI is considered a bag, and the obtained patches are treated as instances. The objective of FSWC is to classify both bags and instances with only a limited number of labeled bags. Unlike conventional few-shot learning problems, FSWC poses additional challenges due to its weak bag labels within the MIL framework. Drawing inspiration from the recent achievements of vision-language models (V-L models) in downstream few-shot classification tasks, we propose a two-level prompt learning MIL framework tailored for pathology, incorporating language prior knowledge. Specifically, we leverage CLIP to extract instance features for each patch, and introduce a prompt-guided pooling strategy to aggregate these instance features into a bag feature. Subsequently, we employ a small number of labeled bags to facilitate few-shot prompt learning based on the bag features. Our approach incorporates the utilization of GPT-4 in a question-and-answer mode to obtain language prior knowledge at both the instance and bag levels, which are then integrated into the instance and bag level language prompts. Additionally, a learnable component of the language prompts is trained using the available few-shot labeled data. We conduct extensive experiments on three real WSI datasets encompassing breast cancer, lung cancer, and cervical cancer, demonstrating the notable performance of the proposed method in bag and instance classification. Codes will be available at `https://github.com/miccaiif/TOP`.

## 1 Introduction

The automated analysis of pathology Whole Slide Images (WSIs) plays a crucial role in contemporary cancer diagnosis and the prediction of treatment response [34, 44, 28, 31, 26, 15, 43, 3]. Unlike natural images, WSIs typically possess a gigapixel resolution, rendering them unsuitable as direct inputs for deep learning models. To address this problem, a common approach involves dividing WSIs into non-overlapping small patches for subsequent processing. However, due to the vast number of patches within a single WSI, it is impractical to assign fine-grained labels to these small patches, rendering instance-level supervised methods unfeasible [39, 9, 7, 27, 29]. Consequently, Multiple Instance Learning (MIL), a popular weakly supervised learning paradigm, has emerged as an effective solution to overcome these challenges. In the MIL framework, each WSI is considered a "bag", and

---

*Corresponding Authors.

37th Conference on Neural Information Processing Systems (NeurIPS 2023).

the extracted patches are regarded as instances within this bag. In a positive bag, there exists at least one positive instance, while in a negative bag, all instances are negative. During training, only the bag labels are known, whereas the instance labels remain unknown [35, 49]. Deep learning-based WSI classification typically involves two tasks: bag-level classification, accurately predicting the category of the target bag, and instance-level classification, accurately identifying positive instances within positively labeled bags.

MIL methods for WSI classification can be broadly categorized into instance-based methods [4, 10, 35, 23] and bag-based methods [18, 41, 22, 56, 40, 8, 29, 48, 46]. Instance-based approaches involve training an instance classifier using artificially-generated pseudo labels to estimate the probability of each instance being positive. These individual predictions are then aggregated to obtain the bag-level prediction. On the other hand, bag-based methods have emerged as the predominant approach for WSI classification. These methods initially extract features for each instance and subsequently employ an aggregation function to combine the features of all instances within a bag into a single bag feature. Finally, a bag classifier is trained using the known labels of the bags. Recently, attention-based aggregation methods [18, 16, 61, 54, 41, 22, 56, 29] have demonstrated promising performance, and they could leverage attention scores assigned to each instance for instance-level classification.

Most existing MIL methods for WSI classification assume the availability of a substantial amount of labeled data at the bag level. However, in clinical practice, limitations such as patient privacy concerns, challenges in obtaining pathological samples, or the diagnosis of rare or emerging diseases often result in a scarcity of pathological data [34, 44, 39, 9, 21]. Consequently, existing methods are ill-equipped to handle such few-shot learning scenarios. In this paper, *we present a novel WSI classification problem termed Few-shot Weakly Supervised WSI Classification (FSWC).* Traditional few-shot learning strives to achieve good classification performance with very few labeled support samples per class (usually only 1, 2, 4, 8, or 16 samples). Similarly, in FSWC, only a few bags are labeled for training (only 1, 2, 4, 8, or 16 per class). Notably, FSWC diverges from traditional few-shot learning on natural images due to the absence of instance-level labels, with only bag-level labels provided. The MIL setting and the absence of instance labels make FSWC considerably more challenging than traditional few-shot learning problems. *The primary objective of FSWC is to achieve precise bag-level and instance-level classification with very few training bags.* Figure 1 A and B provide intuitive illustrations of the existing WSI classification and FSWC tasks.

Recently, significant advancements have been made in visual representation and transfer learning with the emergence of vision-language models (V-L models) such as CLIP [38], ALIGN [20], and FLIP [55]. These models have demonstrated remarkable success, indicating their ability to learn universal visual representations and perform effectively in zero-shot or few-shot settings for downstream tasks. *Motivated by these achievements, we apply V-L models to address the FSWC problem.* Unlike traditional visual frameworks, V-L models employ a two-tower architecture consisting of an Image Encoder and a Text Encoder for pre-training on extensive image-text pairs. The objective is to align images and their corresponding description texts within a shared feature space. For zero-shot classification tasks, a carefully designed prompt template, such as "a photo of a [CLS]," is utilized to classify the target image through similarity matching in the corresponding feature space. To enhance transfer performance and eliminate the need for handcrafted prompt templates, methods like CoOp [60] replace manual prompts with learned prompt representations. These approaches adapt V-L models for few-shot image recognition tasks using a small number of labeled images from the target dataset, with only prompt parameters being trained while the V-L model parameters remain fixed. Figure 1 C provides an intuitive depiction of the fine-tuning paradigm based on pre-trained V-L models and prompt learning in the context of natural images. A straightforward application of V-L models in FSWC involves using the Image Encoder to extract instance features within each bag and aggregating these instance features into bag-level representations using established aggregation functions. Subsequently, prompt learning algorithms like CoOp [60] can be employed at the bag level. However, our experimental results demonstrate unsatisfactory performance with this approach. *The main challenges lie in the absence of efficient instance aggregation methods and the complexity of designing appropriate bag-level prompts.*

In order to effectively utilize V-L models for addressing the FSWC problem, we present a **TwO**-level **P**rompt Learning MIL framework guided by pathology language prior knowledge, referred to as **TOP**. The main concept of TOP is illustrated in Figure 1 D. Initially, we employ the Image Encoder of V-L models to extract instance features within each bag. Subsequently, we introduce prompt guided pooling as a means to aggregate these instance features into a bag-level feature. To facilitate this

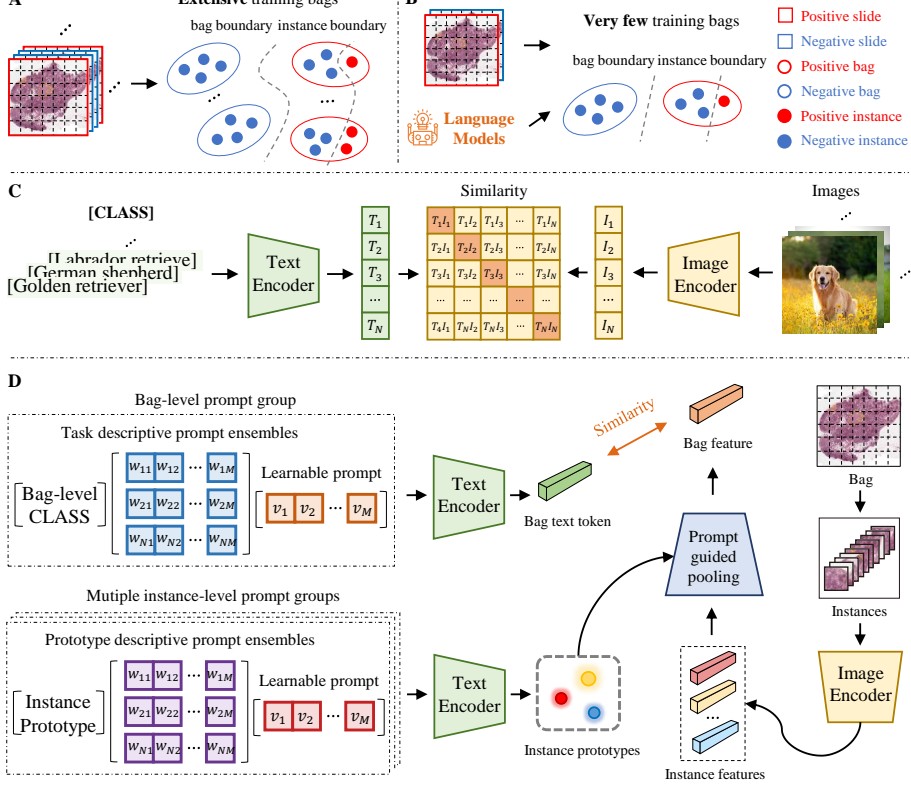

Figure 1: A. Existing WSI classification tasks. B. Our proposed FSWC task based on language models. C. Existing prompt learning paradigm using pre-trained V-L models, where the parameters of the V-L models are always frozen. D. Our proposed Two-level Prompt Learning paradigm.

aggregation process, we leverage the capabilities of the large language model GPT-4 [33] to generate multiple instance-level prompt groups. These prompt groups serve as instance-level pathology language prior knowledge, effectively guiding the aggregation process. Furthermore, GPT-4 [33] is utilized to construct a bag-level prompt group, which is matched with the bag feature to facilitate few-shot prompt learning at the bag level. Throughout this process, the bag-level prompt group acts as pathology language prior knowledge, providing guidance for the few-shot prompt learning process. In TOP, both the instance-level and bag-level prompts comprise three components. The first component encompasses the task label at the instance or bag level, such as "an image patch of [Lymphocytes]" and "a WSI of [Lung adenocarcinoma]." The second component consists of a combination of various visual description texts associated with the given task label. It is important to note that these visual descriptions are not manually designed, but are obtained through a question-answering approach employing GPT-4. For example, we generate prompts such as "What are the visual pathological forms of Lung adenocarcinoma?". Our experimental findings consistently demonstrate the criticality of incorporating task-specific visual descriptions tailored to the pathological WSI classification task. Inspired by CoOp [60], we design the third component as a learnable continuous prompt representation, enabling automatic adaptation and further enhancing transfer performance.

**The main contributions of this paper are as follows:**

• We proposed and effectively solved the novel Few-shot Weakly Supervised WSI Classification (FSWC) problem.

• We propose a Two-level Prompt Learning MIL framework, referred to as TOP. At the instance level, we leverage pathology language prior knowledge derived from GPT-4 to guide the aggregation of instance features into bag features. In addition, at the bag level, we create a comprehensive bag-level prompt group by incorporating bag-level pathology categories and visual pathology descriptions as prior knowledge to facilitate few-shot learning under the supervision of bag labels.

• We conducted comprehensive evaluations on three real-world WSI datasets, including breast cancer, lung cancer, and cervical cancer. TOP demonstrates strong bag classification and instance classification performance under limited bag-level labeled data, achieving state-of-the-art results.

## 2   Related Work

### 2.1   Multiple Instance Learning for WSI Classification

Most MIL methods for WSI classification [18, 54, 41, 22, 56, 40, 8, 48, 24, 46, 19, 37] follow a two-step process: they extract features for each instance and then aggregate these instance features to obtain a representative bag feature. Subsequently, a bag classifier is trained using the bag features and corresponding labels. Notably, attention-based aggregation methods [18, 16, 61, 54, 41, 22, 56, 29] have demonstrated promising results, where attention scores assigned to each instance contribute to instance-level classification. However, these approaches heavily rely on a substantial number of labeled bags, which is not the case in our proposed FSWC task, where only a limited number of labeled bags are available for training. While some recent studies [22, 56, 40, 36, 5, 6, 30] employed pre-trained networks to extract instance features, these models were solely pre-trained on self-supervised learning or ImageNet data. In contrast, we explore the utilization of pre-trained V-L models (as detailed in Section 2.3) for extracting instance features. Additionally, we present a Two-level Prompt Learning MIL framework guided by pathology language prior knowledge.

### 2.2   Few Shot Classification

The primary objective of Few-shot Classification (FSC) is to accurately classify test samples by utilizing limited labeled support examples. Typically, only 1, 2, 4, 8, or 16 examples per category are available. This classification process leverages learned knowledge and prior information [12, 13, 1, 42, 45, 47]. Recently, vision-language models (referred to as V-L models, detailed in Section 2.3) such as CLIP [38], ALIGN [20], and FLIP [55] have demonstrated significant success in FSC. This success suggests that these large models have acquired universal visual representations and exhibit improved performance in downstream tasks, particularly in zero-shot or few-shot scenarios. Nevertheless, the task of few-shot WSI classification under bag-level supervision has not yet been investigated. In this paper, we introduce this paradigm as few-shot weakly-supervised WSI classification (FSWC) and propose an effective solution by employing CLIP-based prompt learning.

### 2.3   Vision-language Models and Prompt Learning

Pre-trained Vision-Language (V-L) models, such as CLIP [38], ALIGN [20], and FLIP [55], which have been trained on extensive image-text pairs, exhibit remarkable potential in visual representation and transfer learning [57, 60, 59, 11, 50, 51, 14, 52, 53, 58]. These V-L models employ a dual-tower architecture that comprises visual and text encoders. They utilize contrastive learning to align text-to-image and image-to-text in the feature space. The pre-trained V-L models, including CLIP [38], demonstrate remarkable transferability in image recognition. By carefully designing text descriptions, referred to as "prompts," to align with the corresponding image features in the feature space, these models enable zero-shot or few-shot classification. Building on the accomplishments of CLIP, CoOp [60] replaces manually created prompts with a learned prompt representation and adapts V-L models to downstream FSC tasks. Motivated by the triumph of V-L models in FSC within the domain of natural images, we propose several techniques to effectively adapt pre-trained V-L models for addressing the FSWC problem.

## 3   Preliminaries

### 3.1   Problem Formulation

Given a dataset $X = \{X_1, X_2, \ldots, X_N\}$ comprising $N$ WSIs, and each WSI $X_i$ is partitioned into non-overlapping small patches $\{x_{i,j}, j = 1, 2, \ldots n_i\}$, where $n_i$ represents the number of patches obtained from $X_i$. All patches within $X_i$ collectively form a bag, and each patch serves as an instance of that bag. The bag is assigned a label $Y_i \in \{0, 1\}$, where $i = \{1, 2, ...N\}$. The labels of each

instance $\{y_{i,j}, j = 1, 2, \ldots n_i\}$ are associated with the bag label in the following manner:

$$Y_i = \left\{ \begin{array}{ll} 0, & \text{if } \sum_j y_{i,j} = 0 \\ 1, & \text{else} \end{array} \right. \tag{1}$$

This implies that all instances within negative bags are assigned negative labels, whereas positive bags contain at least one positive-labeled instance. In the context of weakly-supervised MIL, only the bag label is provided for the training set, while the labels of individual instances remain unknown. The Few-shot Weakly-supervised WSI Classification (FSWC) task poses an even greater challenge as it allows only a limited number of labeled bags for training. In the FSWC task, "shot" refers to the number of labeled slides. In N-shot experiments, the training set uses only N pairs of positive and negative slides for training, and the model is evaluated on the complete testing set. Typically, only a small number of bags per class, such as 1, 2, 4, 8, or 16, are available for training. The objective of FSWC is to accurately classify both the bags and individual instances, despite the scarcity of labeled training bags.

### 3.2 Vision-Language Pre-training and Few-shot Prompt Learning

We first provide a concise overview of the pre-training of V-L models and few-shot prompt learning. In this paper, we use the V-L model of CLIP [38], but our method is also applicable to other CLIP-like V-L models.

**Model and Pre-training.** The CLIP framework comprises an image encoder and a text encoder. The image encoder employs ResNet-50 or ViT to extract image features, while the text encoder uses Transformer to generate text features. CLIP's primary training objective is to establish an embedding space that align image features with their corresponding text features by means of a contrastive loss. During training, a batch of image-text pairs is used, and CLIP maximizes the cosine similarity of matching pairs while minimizing the cosine similarity of all other non-matching pairs. To facilitate the acquisition of diverse visual concepts that can be readily applied to downstream tasks, CLIP is trained on a large-scale dataset consisting of 400 million image-text pairs, which includes medical data.

**Zero-shot Inference.** CLIP has the inherent ability to perform zero-shot classification because it is pre-trained to predict whether an image matches a given text description. Specifically, the approach involves using the image encoder to extract features from the image to be classified, using the text encoder to extract features of the text descriptions of all candidate categories (referred to as "prompts"), and then calculating the degree of match between the image features and all the text features to determine the classification category. Formally, let $z$ be the image feature extracted by the image encoder for image $x$, and let $\{w_i\}_{i=1}^{K}$ be a set of weight vectors generated by the text encoder. Here, $K$ represents the number of candidate categories, and each $w_i$ comes from the prompt "a photo of a [CLASS]", where the CLASS token is replaced with a specific class name, such as "cat", "dog", or "car". The predicted probability of each category is calculated using Equation 2:

$$p(y = i \mid x) = \frac{\exp\left(\cos\left(w_i, f\right)/\tau\right)}{\sum_{j=1}^{K} \exp\left(\cos\left(w_j, f\right)/\tau\right)} \tag{2}$$

Here, $\tau$ is the temperature coefficient learned by CLIP, and $\cos(\cdot, \cdot)$ represents cosine similarity.

**Few-shot Prompt Learning.** Research has shown that the construction of prompts plays a crucial role in the downstream task classification performance. CoOp [60] learns an end-to-end continuous vector as a supplementary prompt using limited labeled data from the downstream task. Formally, the overall prompt is designed as:

$$t = [V]_1[V]_2 \ldots [V]_M[CLASS] \tag{3}$$

where each $[V]_m (m \in \{1, \ldots, M\})$ is a vector of the same dimension as the word embedding (i.e., 512 for CLIP), and $M$ is a hyperparameter that specifies the number of context tokens. The text encoder of CLIP is used to encode the prompt $t$ to obtain a classification weight vector $\{w_i\}_{i=1}^{K}$ that represents the visual concept, and then the prediction probability of each category is calculated according to Equation 2. A small amount of labeled data in the downstream task is used to train

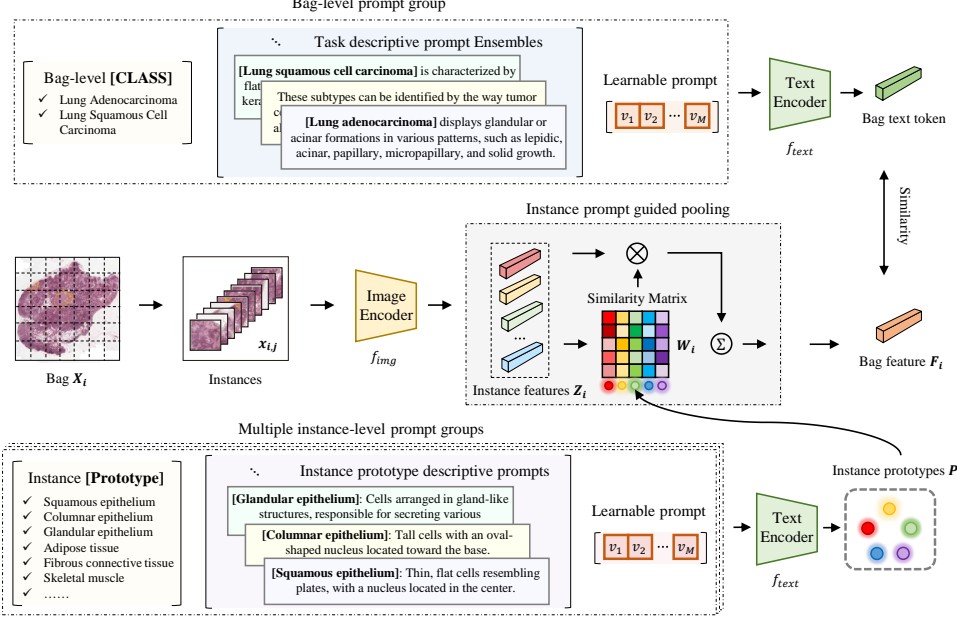

Figure 2: Framework of our proposed TOP.

and optimize $[V]_m$ in the prompt using cross-entropy loss. Other parameters, including the Image Encoder and Text Encoder of the V-L model, are frozen.

$$\text{Loss } = CE(y, p) \tag{4}$$

where $y$ represents the ground truth label and $p$ represents the predicted probability of each class based on the prompt.

## 4 Method

Figure 2 presents our proposed TOP framework. First, we use the Image Encoder $f_{img}$ of CLIP to extract instance features $Z_i$ for all instances within a bag $X_i$. Then, we propose instance prompt guided pooling to aggregate these instance features into a bag feature $F_i$. During this process, we use a large language model GPT-4 to generate multiple instance-level prompt groups and input them into the Text Encoder $f_{text}$ of CLIP to generate instance prototypes $P$, which serve as instance-level language priors to guide the aggregation process. Next, we use GPT-4 to construct a bag-level prompt group and input it into the Text Encoder $f_{text}$ to generate a Bag text token $B_i$. We then match $B_i$ with the aggregated bag feature $F_i$ to complete bag-level few-shot prompt learning. During this process, the bag-level prompt group serves as a bag-level language prior to guide the few-shot prompt learning process. The loss function for few-shot prompt learning is shown in equation 4 in Section 3.2, and the overall training objective is to use a small amount of bag-level labeled data to optimize the learnable prompt vectors $[V]_m$ in the bag-level and instance-level prompt groups (the two $[V]_m$ vectors are different). In addition, during training, we also constrain the minimum correlation between each Instance prototype in $P$ to prevent all prototypes from being too similar and causing degradation. The construction of instance and bag-level prompts will be introduced in Section 4.1 and the instance prompt guided pooling method will be introduced in Section 4.2.

During inference, for bag classification, we calculate the matching degree between the image features and all target class bag prompt features to determine the classification category, as shown in equation 2 in Section 3.2. For instance classification, we obtain the classification score of each instance by averaging the similarity weights established between each instance feature and multiple text-based instance prototypes.

### 4.1 Construction of Instance and Bag-level Prompt

We utilize GPT-4 to construct instance and bag-level prompts as efficient language priors to guide the instance-level feature aggregation and the bag-level few-shot prompt learning.

Instance-level prompt groups are designed to generate visual descriptions of various instance phenotypes as prior knowledge to effectively guide instance-level feature aggregation. Each prompt group corresponds to a potential tissue phenotype that may appear in WSI and consists of three parts. The first part is a text description of various instance phenotypes in the form of "an image patch of [Lymphocytes]". Typically, instances in a WSI contain information about different phenotypes, such as different forms of tumor cells, lymphocytes, epithelial cells, etc. For the second part, we used a question-answering mode with GPT-4 to obtain common visual descriptions of different instance phenotypes in WSIs. Note that these visual descriptions do not require manual design but are obtained from GPT-4, as shown in the Supplementary Materials. For each instance phenotype, we focus on guiding GPT-4 to describe the visual characteristics it has from a visual perspective, thereby establishing visual priors for these phenotypes. Inspired by the learnable prompt in CoOp [60], we design the third part as a learnable prompt representation.

The bag-level prompt group is designed to guide the few-shot prompt learning process at the bag level. It also consists of three parts. The first part is the description of the task label at the bag level in the form of "a WSI of [Lung adenocarcinoma]". The second part is a combination of various visual descriptions for this task label, which are also obtained from GPT-4, as shown in the Supplementary Materials. For each classification task, we guide the GPT-4 model to describe the complex medical concept from a visual perspective, so as to establish a visual prior for the complex medical concept from the textual description. The third part is also designed as a learnable prompt representation.

Detailed bag-level and instance-level task descriptive prompt ensembles are presented in the Supplementary Materials and will be fully open-source.

## 4.2 Instance Prompt Guided Pooling

We propose a prompt guided pooling strategy to aggregate instance features into bag features. The main idea is to first calculate similarity weights between the image features of each instance and the prototypes of multiple text descriptions, and then use the weighted average of all instance features in a bag as the bag feature.

Mathematically, assuming that $\boldsymbol{X_i}$ represents the current bag containing $n_i$ instances, we use the Image Encoder $f_{img}$ to extract instance features $\boldsymbol{Z_i} \in \boldsymbol{R}^{n_i \times m}$, where $m$ represents the dimension of the features.

$$\boldsymbol{Z_i} = f_{img}\left(\boldsymbol{X_i}\right) \tag{5}$$

Then, we input multiple instance-level prompt groups $\boldsymbol{T}$ into the Text Encoder $f_{text}$ to obtain multiple instance prototypes, $\boldsymbol{P} \in \mathbb{R}^{n_p \times m}$, where $n_p$ represents the number of prototypes, which corresponds to the number of instance-level prompt groups. It varies depending on the specific classification task.

$$\boldsymbol{P} = f_{\text{text}}\left(\boldsymbol{T}\right) \tag{6}$$

Next, we calculate the dot product of instance features $\boldsymbol{Z_i}$ and a set of prototypes $\boldsymbol{P}$, and then perform softmax normalization by column (each column corresponds to a prototype), obtaining aggregation weights $\boldsymbol{W_i} \in \mathbb{R}^{n_i \times n_p}$ for each prototype with respect to the current bag. Next, we use the aggregation weights to obtain a set of weighted features $\boldsymbol{W_i}^\top \cdot \boldsymbol{Z_i} \in \mathbb{R}^{n_p \times m}$, and finally average the weights of all prototypes to obtain the bag feature $\boldsymbol{F_i} \in \mathbb{R}^{1 \times m}$.

$$\boldsymbol{W_i} = \text{Softmax}\left(\boldsymbol{Z_i} \cdot \boldsymbol{P}^\top\right) \tag{7}$$

$$\boldsymbol{F_i} = \text{mean}\left(\boldsymbol{W_i}^\top \cdot \boldsymbol{Z_i}\right) \tag{8}$$

In addition, during training, we also constrain the minimum correlation between each instance prototype in $\boldsymbol{P}$ to avoid degeneration, which means that all prototypes are too similar to each other, with the following loss:

$$\text{Loss} = \min\left(\boldsymbol{W}^\top \times \boldsymbol{W}\right) \tag{9}$$

This auxiliary loss aims to separate instance prototypes learned by each instance prompt, ensuring distinct phenotypes representing WSIs. Crucial instance prototypes for slide classification stand out during aggregation.

Table 1: Performance of bag-level classification on the Camelyon 16 Dataset.

| Method | 16-shot | 8-shot | 4-shot | 2-shot | 1-shot |
|---|---|---|---|---|---|
| Linear-Probe (Mean-pooling) | 0.5816 | 0.5627 | 0.4930 | 0.3235 | 0.3357 |
| Linear-Probe (Max-pooling) | 0.5947 | 0.5814 | 0.5417 | 0.5401 | 0.5268 |
| Linear-Probe (Attention-pooling) | 0.7866 | 0.6738 | 0.6594 | 0.6187 | 0.5312 |
| CoOp (Attention-pooling) | 0.8012 | 0.6746 | 0.6639 | 0.6525 | 0.6462 |
| **Instance+Bag Prompt Learning (Ours)** | **0.8301** | **0.7287** | **0.7151** | **0.6958** | **0.6783** |

Table 2: Performance of instance-level classification on the Camelyon 16 Dataset.

| Method | 16-shot | 8-shot | 4-shot | 2-shot | 1-shot |
|---|---|---|---|---|---|
| Linear-Probe (Attention-pooling) | 0.7508 | 0.6630 | 0.6316 | 0.6177 | 0.5934 |
| CoOp (Attention-pooling) | 0.8567 | 0.6648 | 0.6333 | 0.6221 | 0.5923 |
| **Instance+Bag Prompt Learning (Ours)** | **0.8896** | **0.7200** | **0.7142** | **0.7019** | **0.6938** |

# 5 Experiment

## 5.1 Datasets, Evaluation Metrics and Comparison Methods

We comprehensively evaluated the instance classification and bag classification performance of TOP in FSWC tasks using three real-world datasets of different cancer types from different centers: the Camelyon 16 Dataset [2] for breast cancer, the TCGA-Lung Cancer Dataset[2] for lung cancer, and an in-house Cervical Cancer Dataset for cervical cancer. See Supplementary Material for detailed introductions to the datasets.

For both instance and bag classification, we use Area Under Curve (AUC) as the evaluation metric. However, it should be noted that only the Camelyon 16 Dataset has the true labels for each instance, while the other two datasets only have bag-level labels. Therefore, we evaluate the instance and bag classification performance of each method on the Camelyon 16 Dataset, and only evaluate the bag classification performance on the latter two datasets.

Because existing few-shot learning methods cannot be used in FSWC task, we constructed four baselines based on the current state-of-the-art few-shot learning methods CoOp [60] and Linear Probe [38]: (1) Linear-Probe (Mean pooling), (2) Linear-Probe (Max pooling), (3) Linear-Probe (Attention pooling), and (4) CoOp (Attention pooling). Specifically, we first used CLIP as the image feature extractor to extract all instance features within each bag. Then, we aggregated all instance features within a bag using simple Mean, Max pooling or learnable Attention Pooling [18] to obtain the bag feature. Linear-Probe indicates that we used a linear layer to perform bag-level classification on the aggregated bag feature. CoOp indicates that we used the bag-level label and learnable prompt to perform bag-level classification through prompt learning. We conducted few-shot classification experiments with 1, 2, 4, 8, and 16 labeled bags for each class.

## 5.2 Implementation Details

We used the image encoder and text encoder of CLIP as the feature extractors for both images and text. The number of learnable parameters is empirically set to 10 tokens for both instance and bag prompt and other quantities can be used in practice. During training, we fixed all weights of CLIP and only trained the learnable parameters of bag prompt and instance prompt. For experiments with different shots, we randomly trained the network five times with fixed labeled bags and reported the average performance of each method. All comparative methods utilize the same labeled bags.

## 5.3 Results on the Camelyon 16 Dataset

The bag classification and instance classification performance on the Camelyon 16 dataset are shown in Tables 1 and 2, respectively. It can be seen that TOP achieved the best bag and instance classification performance in all few-shot settings, and significantly outperformed all comparison methods by a large margin. It can be observed that Linear-Probe with Mean/Max pooling can hardly work. Although using trainable attention pooling helps learn the importance of each instance and improves the performance of Linear-Probe, it still has limitations in performance. Prompt learning with fully trainable prompts in CoOp outperforms Linear-Probe. In contrast, our method used a two-level prompt learning paradigm, which achieved the best performance on both bag and instance

---

[2]http://www.cancer.gov/tcga

Table 3: Performance of bag-level classification on the TCGA-Lung Cancer Dataset.

| Method | 16-shot | 8-shot | 4-shot | 2-shot | 1-shot |
|---|---|---|---|---|---|
| Linear-Probe (Mean-pooling) | 0.6022 | 0.5418 | 0.4934 | 0.4908 | 0.4646 |
| Linear-Probe (Max-pooling) | 0.6227 | 0.5547 | 0.5155 | 0.4985 | 0.4876 |
| Linear-Probe (Attention-pooling) | 0.7178 | 0.6539 | 0.6248 | 0.5832 | 0.5713 |
| CoOp (Attention-pooling) | 0.7840 | 0.6824 | 0.6811 | 0.6772 | 0.6801 |
| **Instance+Bag Prompt-Learning (Ours)** | **0.8235** | **0.8059** | **0.7531** | **0.7245** | **0.7123** |

Table 4: Performance of bag-level classification on the Cervical Cancer Dataset.

| Method | 16-shot | 8-shot | 4-shot | 2-shot | 1-shot |
|---|---|---|---|---|---|
| Linear-Probe (Mean-pooling) | 0.6756 | 0.6684 | 0.6593 | 0.6246 | 0.6011 |
| Linear-Probe (Max-pooling) | 0.6322 | 0.6249 | 0.6038 | 0.5884 | 0.5869 |
| Linear-Probe (Attention-pooling) | 0.7345 | 0.7282 | 0.7155 | 0.6873 | 0.6137 |
| CoOp (Attention-pooling) | 0.7565 | 0.7349 | 0.7271 | 0.6927 | 0.6484 |
| **Instance+Bag Prompt-Learning (Ours)** | **0.8189** | **0.8007** | **0.7869** | **0.7618** | **0.7052** |

classification, with an average improvement of 4.2% and 7.0%, respectively, over the second-best method.

## 5.4 Results on the TCGA-Lung Cancer Dataset and the Cervical Cancer Dataset

The results on the TCGA-Lung Cancer Dataset are shown in Table 3. It can be seen that TOP still achieves the best bag classification performance in all few-shot settings, and significantly outperforms all competitors by a large margin, with an average improvement of 6.3% over the second-best method.

The results on the Cervical Cancer Dataset are shown in Table 4. It should be noted that this task is extremely challenging, and even pathologists cannot make direct judgments. TOP displays strongest performance, with an average improvement of 6.3% over the second-best method.

Table 5: Ablation results of bag-level classification on the Camelyon 16 Dataset.

| Method | 16-shot | 8-shot | 4-shot | 2-shot | 1-shot |
|---|---|---|---|---|---|
| Bag Prompt+Attention-pooling | 0.8168 | 0.6980 | 0.6706 | 0.6673 | 0.6483 |
| CoOp+Attention-pooling | 0.8012 | 0.6746 | 0.6639 | 0.6525 | 0.6462 |
| CoOp+Prompt guided pooling | 0.8216 | 0.7079 | 0.6833 | 0.6732 | 0.6699 |
| **Bag Prompt+Prompt guided pooling (Ours)** | **0.8301** | **0.7287** | **0.7151** | **0.6958** | **0.6783** |

Table 6: Ablation results of instance-level classification on the Camelyon 16 Dataset.

| Method | 16-shot | 8-shot | 4-shot | 2-shot | 1-shot |
|---|---|---|---|---|---|
| Bag Prompt+Attention-pooling | 0.8699 | 0.6753 | 0.6425 | 0.6399 | 0.5961 |
| CoOp+Attention-pooling | 0.8567 | 0.6648 | 0.6333 | 0.6221 | 0.5923 |
| CoOp+Prompt guided pooling | 0.8754 | 0.6912 | 0.6707 | 0.6515 | 0.6045 |
| **Bag Prompt+Prompt guided pooling (Ours)** | **0.8896** | **0.7200** | **0.7142** | **0.7019** | **0.6938** |

# 6 Ablation Study

We conducted ablation experiments on the two key components of TOP, the instance prompt guided pooling and bag-level prompt group. Experiments are conducted on the Camelyon 16 Dataset, and the bag classification and instance classification results are shown in Table 5 and Table 6, respectively. In the two tables, "Prompt guided pooling" represents the use of our proposed instance prompt guided aggregation method while "Attention-pooling" represents the use of attention-based method to aggregate instance features into bag features; "Bag Prompt" represents the use of our proposed Bag-level prompt group while "CoOp" represents using only the learnable part and bag labels as prompt at the bag-level prompt learning.

*Effectiveness of instance prompt guided pooling.* In both Table 5 and Table 6, no matter 'Bag Prompt' or 'CoOp' is used for the bag-level prompt learning, 'Prompt guided pooling' consistently results in significantly better performance than 'Attention pooling', which indicates the effectiveness of our proposed Prompt guided pooling strategy for instance feature aggregation.

*Effectiveness of Bag-level prompt group.* In Table 5 and Table 6, no matter what instance feature aggregation method is used, 'Bag-prompt' always outperforms 'CoOp', which indicates that our proposed Task descriptive prompt is better than current SOTA methods of fully learnable prompt.

Table 7: Ablation and sensitivity tests of the auxiliary loss on the Camelyon16 dataset.

| Shot | Loss Weight | 0 | 10 | 25 (**ours**) | 50 |
|---|---|---|---|---|---|
| 1-shot | Bag AUC | 0.6691 | 0.6759 | **0.6783** | 0.6771 |
| | Instance AUC | 0.6837 | 0.6908 | **0.6938** | 0.6913 |
| 2-shot | Bag AUC | 0.6867 | 0.6924 | **0.6958** | 0.6945 |
| | Instance AUC | 0.6924 | 0.7008 | **0.7019** | 0.7011 |
| 4-shot | Bag AUC | 0.7087 | 0.7123 | **0.7151** | 0.7154 |
| | Instance AUC | 0.7033 | 0.7095 | **0.7142** | 0.7107 |
| 8-shot | Bag AUC | 0.7137 | 0.7218 | **0.7287** | 0.7259 |
| | Instance AUC | 0.7005 | 0.7188 | **0.7200** | 0.7243 |
| 16-shot | Bag AUC | 0.8245 | 0.8281 | **0.8301** | 0.8299 |
| | Instance AUC | 0.8834 | 0.8879 | **0.8896** | 0.8894 |

*Effectiveness of Auxiliary Loss.* Ablation and sensitivity tests on the Camelyon16 dataset (Table 7) demonstrate that our method is not highly sensitive to the loss weight, but its addition significantly improves performance compared to not using it.

## 7   Discussion

In this paper, we introduce the novel problem of Few-shot Weakly-supervised WSI Classification (FSWC) for the first time. We proposed a Two-level Prompt Learning MIL framework named TOP to solve the FSWC problem effectively. TOP utilizes GPT-4 to generate both instance-level and bag-level visual descriptions to facilitate instance feature aggregation and bag-level prompt learning. Experiments on three WSI classification tasks shows the high performance of TOP in FSWC tasks.

We carefully consider the validity, rationale, motivation and potential model updates' impact for using GPT-4's knowledge. *For validity and rationale:* We rigorously reviewed pathology knowledge descriptions from GPT-4 with three senior pathologists and found them accurate and detailed. Nori *et al.* [32] supports GPT-4's reliability in producing medical domain knowledge due to its vast medical expertise in training data. *For motivation and importance:* Leveraging GPT-4's knowledge as templates enhances efficiency versus manual design. This approach aligns with the few-shot learning goal, easing pathologist annotation. Manual templates might not cover all aspects; specialized doctors' templates could be needed for varied cancer types/tasks. Additionally, different doctors' descriptions vary, lacking a standardized manual description. By leveraging GPT-4's versatility, our aim is to attain knowledge descriptions for multiple cancer types and tasks while avoiding manual domain biases. *For impact of model updates:* GPT-4's language descriptions contributed to training pathology models in our research. We will publicly share all used descriptions, codes, and models. This disclosure ensures reproducibility in reported tasks without the need of invoking GPT-4 for inference or new training. GPT-4's upgrades won't influence current outcomes. We'll explore if GPT upgrades generate new descriptions and their effect on results.

As widely acknowledged, training a pathological foundation model demands a substantial corpus of correlated pathological images and textual data, in addition to significant computational resources. This poses considerable challenges in the field of computational pathology, characterized by high data privacy, scarce annotations, and extensive storage requirements. More recently, prominent research endeavors [25, 17] have entered the spotlight, aiming to develop a comprehensive vision-language model within the domain of digital pathology. Nevertheless, the focus of this study is to some extent orthogonal to these studies. The objective of our method is to develop new prompt-learning strategies based on an existing large vision-language model for few-shot learning. More importantly, our method can be combined with any large vision-language models (either in general natural image domain or pathology-specific domain) for few-shot WSI classification, not limited to the CLIP model used in the paper.

However, this paper has certain limitations, mainly due to the effectiveness of the proposed prompt depending on the capabilities of the large model. If the large model fails to provide effective descriptions, it can affect the model's performance. This study aims to inspire further research that combines foundational models with large-scale language models for the classification of pathology Whole Slide Images. Such research endeavors will herald a new era in AI pathology.

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
