# Supplementary Material

The supplementary material is divided into three sections. Section 1 provides an introduction to the details of the datasets used in this study. Section 2 provides additional details about the construction of instance and bag-level prompts. Section 3 offers further details about the algorithm stability with different shots.

## 1 More Details of Datasets

We comprehensively evaluated the instance classification and bag classification performance of our TOP in the few-shot weakly supervised Whole Slide Image classification (FSWC) tasks using three real-world datasets of different cancer types from different centers: the Camelyon 16 Dataset [1] for breast cancer, the TCGA-Lung Cancer Dataset [1] for lung cancer, and an in-house Cervical Cancer Dataset for cervical cancer. A detailed description of the datasets is as follows.

### 1.1 Camelyon16 Dataset

The publicly available Camelyon16 dataset is a valuable resource for the detection of breast cancer metastasis in lymph nodes [1]. It comprises 400 H&E-stained Whole Slide Images (WSIs) of lymph nodes, with 270 WSIs allocated for training and 130 for testing. WSIs that exhibit metastasis are labeled as positive, whereas those without metastasis are categorized as negative. Importantly, this dataset provides both slide-level labels indicating the overall positivity or negativity of a WSI, as well as pixel-level labels that precisely identify the specific areas of metastasis.

In the weakly supervised scenario studied in this paper, our training process relied only on slide-level labels. To evaluate the instance classification performance of each algorithm, we utilized pixel-level labels of cancerous regions in the test set. Similar to the preprocessing in [2, 4], prior to the training process, we partitioned each WSI into non-overlapping image patches, each measuring $512 \times 512$ pixels at a $10 \times$ magnification. Patches exhibiting an entropy value lower than 5 were discarded as background noise. Additionally, a patch was considered positive if it encompassed 25% or more cancerous areas; otherwise, it was assigned a negative label. As a result, we obtained a total of 186,604 instances for further analysis.

### 1.2 TCGA Lung Cancer Dataset

The TCGA Lung Cancer dataset, which can be accessed through The Cancer Genome Atlas (TCGA) Data Portal, comprises 1054 WSIs. This dataset specifically targets two distinct subtypes of lung cancer: Lung Adenocarcinoma and Lung Squamous Cell Carcinoma. Our primary goal is to achieve precise diagnoses for both subtypes. In this context, WSIs depicting Lung Adenocarcinoma are categorized as negative, while those showcasing Lung Squamous Cell Carcinoma are classified as positive.

It is worth emphasizing that the provided dataset only includes slide-level labels, and patch-level labels are not accessible. Similar to the preprocessing in [2, 4], at a magnification of $20 \times$, the dataset

---

[1]http://www.cancer.gov/tcga

Submitted to 37th Conference on Neural Information Processing Systems (NeurIPS 2023). Do not distribute.

consists of approximately 5.2 million patches, averaging around 5,000 patches per slide. To facilitate the dataset partitioning, 840 slides were allocated for training, while 210 slides were reserved for testing. It is important to note that four slides exhibiting low quality or corruption were excluded from the dataset.

### 1.3 Cervical Cancer Dataset

The Cervical Cancer dataset comprises an in-house clinical pathology dataset containing 374 H&E-stained WSIs of primary lesions of cervical cancer obtained from distinct patients, subsequent to slide selection. All patients underwent abdominal hysterectomy with pelvic lymph node dissection ± para-aortic lymph node dissection. Following the surgical procedure, the lymph node status of each patient was determined by experienced gynecological pathologists. Moreover, all patients possess meticulous follow-up records spanning over a period exceeding five years, along with the outcomes of major immunohistochemical marker detection. We employed this dataset to accomplish the task of predicting lymph node metastasis from primary lesions, which cannot be directly ascertained by physicians through the examination of H&E-stained slides.

The utilization of deep learning technology holds significant clinical implications in predicting the mentioned clinical task directly from H&E-stained slides. By conducting automated analysis on H&E-stained slides from cervical cancer patients, it becomes feasible to effectively predict lymph node metastasis in primary lesions prior to treatment. This predictive capability aids in the selection of suitable treatment plans and personalized therapies.

Similar to the preprocessing in [4], the experiment was conducted at a magnification of 5×, and each WSI was partitioned into non-overlapping patches of size 224×224 to form a bag. Background patches with entropy values below 5 were excluded from the original WSI. The slides corresponding to patients who experienced pelvic lymph node metastasis were labeled as positive (209 cases), while those who did not develop pelvic lymph node metastasis were labeled as negative (165 cases). The WSIs were randomly divided into a training set (300 cases) and a test set (74 cases).

## 2 Construction of Instance and Bag-level Prompts

We utilize the large language model GPT-4 [3] to construct instance and bag-level prompts as efficient language priors to guide the instance-level feature aggregation and the bag-level few-shot prompt learning.

Instance-level prompt groups are designed to generate visual descriptions of various instance phenotypes as prior knowledge to effectively guide instance-level feature aggregation. Each prompt group corresponds to a potential tissue phenotype that may appear in WSIs and consists of three parts. The first part is a text description of various instance phenotypes in the form of "an image patch of [Lymphocytes]". Typically, instances in a WSI contain information about different phenotypes, such as different forms of tumor cells, lymphocytes, epithelial cells, etc. For the second part, we used a question-answering mode with GPT-4 to obtain common visual descriptions of different instance phenotypes in WSIs. Note that these visual descriptions do not require manual design but are obtained from GPT-4. Representative instance prototype descriptive prompts of different tasks are shown in Figure 1, 2 and 3. Please note that the construction of instance prototype descriptive prompts depends on the specific task. For example, in the tasks of tumor detection (Camelyon16 Dataset) and lung cancer subtyping (TCGA Lung Cancer Dataset), since the input slides have not been annotated to distinguish different regions, there may be various tissue phenotypes other than cancer cells. Therefore, we construct instance prototype descriptive prompts based on the descriptions corresponding to different tissue phenotypes. On the other hand, in the task of determining lymph node metastasis from primary lesions (Cervical Cancer Dataset), as the network input consists of pre-annotated tumor regions, our focus is mainly on different phenotypic information related to tumor cells when constructing instance prototype descriptive prompts. These descriptions provide medical prior information from different perspectives on the factors leading to high and low probability of lymph node metastasis. For each instance phenotype, we focus on guiding GPT-4 to describe the visual characteristics it has from a visual perspective, thereby establishing visual priors for these phenotypes. Inspired by the learnable prompt in CoOp [5], we design the third part as a learnable prompt representation.

Kindly enumerate common tissues in histological specimens and provide comprehensive descriptions of the characteristics of each type of tissue.

Formatting the responses as **{tissue type} - {description of characteristics}** will help to ensure clarity and organization.

**GPT-4** 🤖 ↓

**Epithelial tissue.**
This tissue type is characterized by tightly packed cells that form a continuous layer covering body surfaces, lining cavities, and forming glands. It functions as a barrier, protecting the body from pathogens and water loss, and has roles in secretion, absorption, and sensory reception. Epithelial tissue is divided into simple and stratified types, based on the number of layers, and squamous, cuboidal, and columnar types, based on cell shape.

**Connective tissue.**
Composed of cells, fibers, and an extracellular matrix, connective tissue serves to support, bind, and protect other tissues and organs. It includes a diverse group of tissue types, such as loose connective tissue, dense connective tissue, adipose tissue, cartilage, bone, and blood. Characteristics include varying degrees of vascularity, a range of cell types (e.g., fibroblasts, adipocytes, chondrocytes), and fibers (e.g., collagen, elastin, reticular).

**Muscle tissue.**
This tissue type is characterized by its contractile nature, which enables movement, maintenance of posture, and heat production. Muscle tissue is classified into three types: skeletal, smooth, and cardiac. Skeletal muscle is striated, voluntary, and attaches to bones; smooth muscle is non-striated, involuntary, and found in the walls of hollow organs; and cardiac muscle is striated, involuntary, and found only in the heart.

**Nervous tissue.**
Comprised primarily of neurons and supporting cells called neuroglia, nervous tissue is responsible for transmitting electrical signals and processing information within the body. Neurons possess specialized structures, such as dendrites, axons, and cell bodies, to facilitate communication. Neuroglia provide structural support, insulation, and protection to neurons.

**Lung Adenocarcinoma.**
A subtype of non-small cell lung cancer, lung adenocarcinoma is characterized by the presence of glandular structures, mucin production, and lepidic growth patterns. It often originates in the peripheral regions of the lung and is the most common type of lung cancer in non-smokers.

**Lung Squamous Cell Carcinoma.**
Another subtype of non-small cell lung cancer, lung squamous cell carcinoma arises from the squamous epithelial cells lining the bronchi. It is characterized by the presence of keratinization, intercellular bridges, and large, polygonal-shaped cells. This type of lung cancer is strongly associated with smoking and typically arises in the central regions of the lung.

……

Figure 1: Representative instance prototype descriptive prompts for the tasks of tumor detection (Camelyon16 Dataset) and lung cancer subtyping (TCGA Lung Cancer Dataset). Here, only six typical prompts are listed, but in reality, we utilized a total of 26 prompts.

Please provide a summary of the factors found in primary tumor whole slide images that may indicate {**a higher probability of metastasis**}, along with a description of their image features.

**GPT-4** 🤖 ↓

**Tumor grade:**
The tumor grade is determined by the degree of abnormality of the cancer cells and their arrangement within the tissue. In whole slide images, higher grade tumors display more disorganized and irregular cell patterns, with increased nuclear atypia (abnormal size and shape of cell nuclei) and a higher mitotic rate (increased number of dividing cells).

**Tumor size:**
The size of the primary tumor can be determined by examining the whole slide image. Larger tumors can be seen occupying a more significant portion of the tissue sample and are generally associated with a higher risk of metastasis.

**Lymphovascular invasion (LVI):**
In whole slide images, LVI is observed as cancer cells within the lumens of lymphatic or blood vessels. This can appear as clusters or single cells within the vessel walls or channels, suggesting that cancer cells have penetrated these channels, increasing the likelihood of metastasis.

**Tumor margins:**
Tumor margins refer to the edges of the tumor tissue in the whole slide image. Infiltrative or poorly defined margins can appear as irregular, jagged, or intermingled with surrounding healthy tissue, which may suggest a higher risk of cancer cells breaking away and potentially leading to metastasis.

**Inflammatory response and stromal reaction:**
In whole slide images, an intense inflammatory response can be seen as an accumulation of immune cells (such as lymphocytes) around the tumor, while a strong stromal reaction is characterized by an increased density and disorganization of the connective tissue surrounding the tumor. Both of these features may be associated with a more aggressive tumor and an increased risk of metastasis.

......

Figure 2: Representative instance prototype descriptive prompts for the task of determining lymph node metastasis from primary lesions (Cervical Cancer Dataset). Here, only five typical prompts with a higher probability of metastasis are listed, but in reality, we utilized a total of 26 prompts.

The bag-level prompt group is designed to guide the few-shot prompt learning process at the bag level. It also consists of three parts. The first part is the description of the task label at the bag level in the form of "a WSI of [Lung adenocarcinoma]". The second part is a combination of various visual descriptions for this task label, which are also obtained from GPT-4, as shown in Figure 4 and 5. For each WSI classification task, we guide the GPT-4 model to describe the complex medical concept from a visual perspective, so as to establish a visual prior for the complex medical concept from the textual description. The third part is also designed as a learnable prompt representation.

All detailed bag-level and instance-level task descriptive prompt ensembles will be fully open-source.

## 3 Algorithm Stability with Different Shots

Unlike general few-shot scenarios, few-shot weakly supervised WSI classification is a very challenging setting. When the randomly selected bags have poor representativeness, the network usually struggles to learn meaningful results. Therefore, for experiments with different shots, we randomly trained the network five times with different labeled bags and reported the highest performance of each method. We further demonstrate the stability of our method under different shots. We ran our method five times on 1, 2, 4, 8, and 16 shots, and recorded the Standard Deviation (STD) of bag AUC and instance AUC, as shown in Figure 6. From the figure, it can be observed that the algorithm exhibits instability when there are too few support bags, which is understandable in the

Please provide a summary of the factors found in primary tumor whole slide images that may indicate {**a lower probability of metastasis**}, along with a description of their image features.

**GPT-4** ⬇️

**Well-differentiated tumor cells:**
Well-differentiated tumor cells exhibit a more organized and differentiated appearance compared to poorly differentiated cells. This can be observed through distinct cell borders, regular nuclear shape and size, and a lower nuclear-to-cytoplasmic ratio.

**Low mitotic activity:**
Low mitotic activity refers to a reduced number of cells undergoing division. In whole slide images, this can be identified by a lower density of cells in the mitotic phase, with fewer visible mitotic figures such as condensed chromosomes or spindles.

**Intact basement membrane:**
The presence of an intact basement membrane, which is a thin layer of extracellular matrix surrounding epithelial tissues, is associated with a lower likelihood of metastasis. This can be identified in whole slide images by a continuous and uninterrupted boundary between tumor cells and the surrounding stroma.

**Lack of vascular invasion:**
Vascular invasion refers to the infiltration of tumor cells into blood or lymphatic vessels. The absence of vascular invasion can be observed in whole slide images by the absence of tumor cells within the lumen of blood vessels or lymphatic channels.

**Minimal stromal reaction:**
A minimal stromal reaction indicates limited inflammation or fibrosis in the surrounding tissue. In whole slide images, this can be characterized by a low density of inflammatory cells, absence of fibrotic areas, and limited deposition of extracellular matrix components.

......

Figure 3: Representative instance prototype descriptive prompts for the task of determining lymph node metastasis from primary lesions (Cervical Cancer Dataset). Here, only five typical prompts with a lower probability of metastasis are listed, but in reality, we utilized a total of 26 prompts.

Please provide the characteristics of WSIs of {**Lung Adenocarcinoma**}?

Please provide the characteristics of WSIs of {**Lung Squamous Cell Carcinoma**}?

**GPT-4** ⬇️

**Lung Adenocarcinoma -** A subtype of non-small cell lung cancer, lung adenocarcinoma is characterized by the presence of glandular structures, mucin production, and lepidic growth patterns.

**Lung Squamous Cell Carcinoma -** Another subtype of non-small cell lung cancer. It is characterized by the presence of keratinization, intercellular bridges, and large, polygonal-shaped cells.

Figure 4: Representative bag descriptive prompts for the task of lung cancer subtyping (TCGA Lung Cancer Dataset).

Please provide the characteristics of WSIs with a {**higher risk of metastasis**} from the primary lesion.

Please provide the characteristics of WSIs with a {**lower risk of metastasis**} from the primary lesion.

**GPT-4** 🤖 ↓

**Higher risk of metastasis** can be suggested by image features such as a high tumor grade with abnormal cell patterns, larger tumor size, presence of lymphovascular invasion, infiltrative tumor margins, and intense inflammatory response and stromal reaction.

**Lower risk of metastasis** is characterized by well-organized cell patterns with low nuclear atypia, smaller tumor size, absence of lymphovascular invasion, well-defined tumor margins, and mild inflammatory response and stromal reaction.

Figure 5: Representative bag descriptive prompts for the task of determining lymph node metastasis from primary lesions (Cervical Cancer Dataset).

highly challenging few-shot weakly supervised classification (FSWC) task. However, as the number of shots increases, the performance of our method gradually becomes stable.

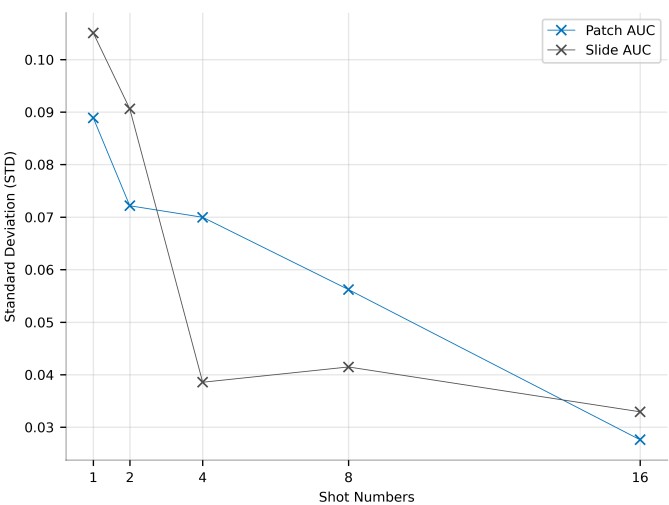

Figure 6: Algorithm stability with different shots.