# OpenReview forum: "The Rise of AI Language Pathologists: Exploring Two-level Prompt Learning for Few-shot Weakly-supervised Whole Slide Image Classification"
_NeurIPS.cc/2023/Conference — NeurIPS 2023 poster_

### Official Review · Reviewer_r2fX · 2023-06-12

**Soundness:** 3 good
**Presentation:** 2 fair
**Contribution:** 2 fair
**Rating:** 4
**Confidence:** 5

**Summary:**

The paper introduces a new approach called Few-Shot Weakly Supervised Learning for Pathology Whole Slide Image classification (FSWC), which aims to classify bags and instances within a WSI with only a limited number of labeled bags. The proposed solution utilizes a large language model, GPT-4, and prompt learning. The approach leverages CLIP to extract instance features for each patch and uses a prompt-guided pooling strategy to aggregate these instance features into a bag feature. The language prior knowledge is obtained using GPT-4 in a question-and-answer mode at both the instance and bag levels. The method is evaluated on three real WSI datasets encompassing breast cancer, lung cancer, and cervical cancer, and demonstrates notable performance in bag and instance classification.

**Strengths:**


The idea is interesting, and is one of the early works trying to apply large language and cross-modal models to WSI (Whole Slide Imaging) image analysis problems. Due to the knowledge learned wit the prompt, requirement on large number of traing samples can thus be alleviated.

**Weaknesses:**

While the idea is interesting, there are significant problems in presentation and languages of the paper. Lots of technical details are not clearly explained. The experiments are not intensive as well, more sota methods need to be compared.

Pls see the section of questions for details.

After reading other reviewer's comments and the rebuttal, I think the paper reauires significant works to improve it, and currently it's not ready to be included in NeurIPS

**Questions:**

1.Unlike general few-shot scenarios，what does shot represent in medical images ? And how is the dataset divided ?
2.In one part of the instance-level prompt, various instance phenotypes are generated. For different datasets, whether the generated functional descriptions are the same or not, without professional knowledge, how to determine these functional descriptions ?
3.What is learnable in prompt learning ?
4.The description of methodology is not clear enough. For example, how does the guided pooling work ?
5.More comparative experiments based on MIL should be added.
6.The reviewer would suggest the author to proofread the manuscript and the quality of the writing and the presentation should be significantly improved.

---

> ### Author Rebuttal · Authors · 2023-08-09
>
> **Q1**. Meaning of Shot and Division of the dataset.
>
> **Response**. In Few-shot WSI classification, "shot" refers to the number of labeled slides. For publicly available datasets, we follow official guidelines to split the data into training (including validation) and testing sets. In-house datasets are randomly divided into training (including validation) and testing sets. In N-shot experiments, the training set uses only N pairs of positive and negative slides for training, and the model is evaluated on the complete testing set. In **Response to Q1 of the General Response**, we have included the results of five trials using randomly selected N slides for training and five trials using fixed N slides for training.
>
> **Q2**. Are generated instance prompts varying from different datasets? how to determine?
>
> **Response**. Instance prototype descriptive prompts vary based on the specific task. For tumor detection and lung cancer subtyping, prompts consider different tissue phenotypes. In the task of determining lymph node metastasis from primary lesions, the focus is on phenotypic information related to tumor cells. Representative prompts for each task are in Supplementary materials Figure 1, 2, and 3.
>
> We rigorously reviewed pathology knowledge descriptions from GPT-4 with three senior pathologists and found them accurate and detailed. Literature [1] supports GPT-4's reliability in producing medical domain knowledge due to its vast medical expertise in training data.
>
> [1] Nori et al. Capabilities of gpt-4 on medical challenge problems. arXiv:2303.13375 (2023).
>
> **Q3**. What is learnable in prompt learning?
>
> **Response**. In our approach, the last 10 tokens in the bag-level and instance-level prompt groups (referred to as learnable prompts) are trainable and treated as learnable parameters. These tokens are encoded into 10 features of 512 dimensions. On the other hand, other descriptive prompts are fixed and directly encoded by the language encoder in CLIP. During the training process, the tokens in the learnable prompts are dynamically learned. For detailed formulas and training methods, please refer to the main text, Section 3.2.
>
> **Q4**. How does the guided pooling work?
>
> **Response**. Prompt guided pooling (PGP) is designed for few-shot WSI classification (FSWC). The key challenge in MIL is aggregating instance features critical for slide classification. In FSWC, limited supervised information hinders the effectiveness of max/mean pooling methods in aggregating crucial instances. Using a separate attention module may lead to overfitting. PGP utilizes GPT-4 to generate visual descriptions of instance phenotypes, guiding feature aggregation. It calculates similarity weights between instance features and text prototypes, then uses the weighted average of all instance features as the bag feature.
>
> **Q5**. More comparative experiments based on MIL should be added.
>
> **Response**. Limited research exists on prompt learning+MIL for Few-Shot Weakly Supervised WSI Classification (FSWC), and methods available for direct comparison are quite limited. We constructed four baselines based on the SOTA few-shot learning methods CoOp and Linear Probe: Linear-Probe+Mean/Max/Attention pooling, and CoOp+Attention pooling. More comprehensive experiments are supplemented in the rebuttal and please see **Response to Q1 and Q2 of General Response** and **Response to Q2 of Reviewer 5SeZ**. In future work, we aim to compare more MIL methods under the FSWC setting to advance research in this area.
>
> **Q6**. Improve the quality and presentation.
>
> **Response**. We will further improve the paper in the camera-ready version.

---

> > ### Comment · Reviewer_r2fX · 2023-08-18
> >
> > Thanks for the rebuttal, which help clarify some of the concerns. But I still think there are lots of works to be done, so I keep my rating

---

> > > ### Author Response · Authors · 2023-08-19
> > >
> > > Thank you very much for your comments and we are happy that our response clarifies some of your concerns. As a major part of work left to be done after our first round of response, we finished experiments on *the TCGA* and *Cervical Cancer datasets* using **non-fixed** and **fixed** bags. We report the Mean and STD of five runs in our new comment “**Mean AUC and STD on the TCGA and Cervical Cancer datasets**” to our **General Response** at the beginning.

---

### Official Review · Reviewer_7zsM · 2023-07-03

**Soundness:** 2 fair
**Presentation:** 3 good
**Contribution:** 3 good
**Rating:** 6
**Confidence:** 4

**Summary:**

This work proposes TOP - a framework using Vision-Language models for few shot Multiple Instance Learning on pathology datasets. TOP uses features generated by passing natural language prompts through a LLM to guide the pooling process for the bag level aggregation in MIL. Results show improvements over other few shot MIL approaches in 3 pathology datasets.

**Strengths:**

- Few shot learning for MIL in histopathology is important since the effective number of data points is greatly reduced when predicting at WSI-level instead of patch-level. This paper thus addresses this highly relevant problem by providing bag level predictions with only a few training bags.
- The prompt-guided pooling idea, where similarity of natural language embeddings of text descriptions of different kinds of pathology to image embeddings is used to ground and calculate the instance-level weights, makes a lot of intuitive sense.

**Weaknesses:**

- My main concern pertains to section 3 in supplementary material, where it is mentioned that "..we randomly trained the network five times with different labeled bags and reported the highest performance of each method..". This implies that the performance comparison is biased towards TOP.
- There can be additional ablations around using pathology specific LLMs [1] instead of generic ones to check if this improves the large variance in performance for TOP across lower number of shots
- The motivation behind adding the correlation loss between the instance prototypes can be clarified further. Also, details around the sensitivity of the model performance to the relative weight of this loss should be shared.

[1] - Santos et al, PathologyBERT - Pre-trained Vs. A New Transformer Language Model for Pathology Domain



**Questions:**

- For comparison on instance-level predictions, [1] uses the Additive MIL framework to get exact instance-level classwise predictions. Can this method be added to the comparison?
- Any specific reason for restricting the learnable params to 10 tokens for instance and bag prompts?

**Limitations:**

- Can the authors give a ballpark estimate of the cost associated with using LLMs for the description generation as a function of the number of prototypes?

---

> ### Author Rebuttal · Authors · 2023-08-09
>
> **Q1**. performance comparison is biased.
>
> **Response**.
>
> Firstly, "training five times with non-fixed labeled bags" in the original text means random selection of different labeled bags in each run based on shot count to compose the training set. The test set is pre-divided and remains unseen during training. All comparative methods use the same training and test sets in each run, ensuring fair comparisons.
>
> In the few-shot learning setup, limited labeled bags for training can cause performance disparities due to varying bag representativeness (like selecting slides with larger or smaller tumor areas). This leads to high standard deviation (STD) across runs with different labeled bags for training. This work is one of the first few-shot WSI learning studies, and there is not a relevant benchmark. Therefore, we conducted five randomized trainings with non-fixed labeled bags, studying the algorithm's holistic performance across diverse slide representativeness. As suggested by multiple reviewers, we supplemented five runs on Camelyon16 dataset using non-fixed bags, reporting Mean AUC (left panel) and STD (right panel) in Table 1 (bag AUC) and Table 2 (instance AUC). As the shot count increases, standard deviations of all methods tend to stabilize. Our proposed method still achieves the best performance. Mean AUC and STD on the other datasets will be provided in a camera-ready version.
>
> Furthermore, we supplemented five rounds of training using fixed labeled bags, employing the same bag in each run for training. Mean AUC (left panel) and STD (right panel) are presented in Table 3 (bag AUC) and Table 4 (instance AUC). The consistent labeled bags notably decrease variability and our method still achieves the best performance.
>
> **Q2**. Ablations using pathology specific LLMs [1] instead of GPT-4’s ones.
>
> **Response**.
>
> We thoroughly examined the model in [1], but it appears restricted to masked language modeling and lacks question-answering abilities like GPT-4. Nevertheless, the concept of using pathology-specific LLMs to generate domain-tailored knowledge shows promise. We intend to investigate this approach further in our future research.
>
> **Q3**. The motivation, ablation and sensitivity tests of auxiliary loss.
>
> **Response**.
>
> The auxiliary loss aims to separate instance prototypes learned by each instance prompt, ensuring distinct phenotypes representing WSIs. Crucial instance prototypes for slide classification stand out during aggregation. New ablation and sensitivity tests on the Camelyon16 dataset (Table 5) demonstrate that our method is not highly sensitive to the loss weight, but its addition significantly improves performance compared to not using it.
>
> **Q4**. Add Additive MIL [1] to the comparison.
>
> **Response**.
>
> Our prompt learning framework can be combined with any bag-level and instance-level MIL methods, including the Additive MIL [1] mentioned. We will include this experiment in the camera-ready version of our work.
>
> **Q5**. Reason for restricting the learnable params to 10 tokens.
>
> **Response**.
>
> Token selection is based on empirical choices, and other quantities can be used in practice. For a comparison of parameter size experiments, please see **Response to Q5 of reviewer 5SeZ**.
>
> **Q6**. Ballpark estimate of the cost using LLMs for generation.
>
> **Response**.
>
> We employed GPT-4's question-answering mode to obtain slide-level and instance-level knowledge descriptions. The generation of prompts for a single task takes around 2 minutes, and experienced pathologists spend approximately 10-20 minutes organizing and reviewing these prompts. These descriptions do not need to be regenerated during inference.

---

> > ### Comment · Reviewer_7zsM · 2023-08-12
> >
> > - Thank you for addressing the comments and sharing the additional results pertaining to training with fixed labeled bags. Its encouraging to see that TOP consistently outperforms other baselines in this evaluation scheme.
> > - The authors have shared ablations around impact of auxiliary loss' weight in Table 5.
> > - Thank you for including experiments with instance-level MIL (Additive MIL) [1] in the final version.
> > After going through the new data from the rebuttal response, I improve my original rating.
> >
> > [1] - S A Javed et al, "Additive MIL: Intrinsically Interpretable Multiple Instance Learning for Pathology"

---

> > > ### Author Response · Authors · 2023-08-12
> > > **Gratitude for Your Review**
> > >
> > > Thank you for reviewing our paper. We highly appreciate your valuable comments, which unquestionably elevate our research quality. We eagerly anticipate incorporating the "Additive MIL" comparison results into the final version.

---

### Official Review · Reviewer_5SeZ · 2023-07-07

**Soundness:** 1 poor
**Presentation:** 3 good
**Contribution:** 3 good
**Rating:** 4
**Confidence:** 4

**Summary:**

The authors introduce a two-level prompt learning framework for label-efficient classification of WSIs using pretrained visual language encoders. The main novelty of the proposed method seems to be that at the instance-level, domain-knowledge + GPT-4 guided prompt prototype groups are used to guide pooling of instance embeddings into the slide-level embedding and while the use of learnable prefix tokens + pretrained text embeddings for few-shot classification follows closely of the established V+L few-shot literature. Experimentation was performed on several WSI datasets (CAMELYON16, TCGA, and an in-house cervical cancer dataset).


**Strengths:**

- This work explores several interesting ideas regarding instance- and bag-level prompts. Previous and concurrent works (e.g. - MI-Zero, PLIP [1,2]) have only explored the application of "instance-level" prompts (text prompts correlated with pathology region-of-interests) for computational pathology. The application of using bag-level prompts in combination of with GPT-4 provides new mechanisms for training and evaluating models for slide-level tasks in pathology.

1. Lu et al. 2023, Visual Language Pretrained Multiple Instance Zero-Shot Transfer for Histopathology Images. CVPR 2023.
2. Huang et al. 2023, Leveraging medical Twitter to build a visual–language foundation model for pathology AI. bioRxiv 2023.

**Weaknesses:**

While the study presents interesting ideas which the reviewer believes to be valuable to the machine learning and computational pathology community. the reviewer has several concerns regarding the evaluation framework and ablation studies.

1. In the methodological details of this work, because the results in the few-shot settings can be highly variable, therefore for each shot, 5 randomly sampled sets of training bags are used, and the highest performance for each model is reported. The reviewer find this practice somewhat questionable given that stability/consistency of the algorithm should be an important factor in deciding which algorithms works well in the few-shot setting. By only reporting the highest performing run on the test set, we can have the scenario where Algorithm 1 obtains consistently reasonable, above chance performance (e.g. [0.7, 0.7, 0.7, 0.7, 0.7]) while Algorithm 2 obtains performance below chance but a single lucky run (e.g. [0.4, 0.4, 0.4, 0.4, 0.75]) and the conclusion will be that Algorithm 2 outperforms Algorithm 1 by a wide margin of 5% - which is not a reasonable conclusion. In the few-shot setting reported in the paper, if the SD of the 5 runs can be as high as 10% - how can the reviewer be certain the proposed model in fact does outperform the other methods consistently when only the highest performing run is reported? I would encourage this work to instead use the median for comparing the different models instead of the max, which is more robust to outliers compared to the mean. Additionally, the exact numbers of all 5-runs should be reported as a box plot in the supplement, and other metrics besides AUC (e.g. balanced accuracy) should be reported as well.

2. This work does not make comprehensive comparison to other methods/in the ablation experiments. An important concern being that the proposed method uses both learnable prefix tokens at the instance level and at the slide-level compared to CoOP, therefore effectively doubling the number of learnable parameters. (a) What happens when the proposed method uses the same number of learnable parameters as CoOP? (b) What happens if we cut the number of prefix tokens in half at both the slide-level and the instance-level relative to CoOP such that the total parameter count is the same?

3. Some design choices in the proposed method do not seem to be properly ablated, for instance. During training, an auxiliary loss (equation 9) is used to encourage de-correlation of different prompt prototypes, but the effectiveness of this choice does not seem to be ablated.

**Questions:**

Summarizing the concerns above:
1. What is the performance when using the median versus the max?
2. Effect of reducing the # of prefix tokens so that the total # of parameters is the same?
3. Ablation study concerning the auxiliary loss?

**Limitations:**

- Limitations discussed in this work include the effectiveness of the prompt depending on the quality of visual representations.

---

> ### Author Rebuttal · Authors · 2023-08-09
>
> **Q1**. Using the median versus the max.
>
> **Response**.
>
> Firstly, "training five times with non-fixed labeled bags" in the original text means random selection of different labeled bags in each run based on shot count to compose the training set. The test set is pre-divided and remains unseen during training. All comparative methods use the same training and test sets in each run, ensuring fair comparisons.
>
> In the few-shot learning setup, limited labeled bags for training can cause performance disparities due to varying bag representativeness (like selecting slides with larger or smaller tumor areas). This leads to high standard deviation (STD) across runs with different labeled bags for training. This work is one of the first few-shot WSI learning studies, and there is not a relevant benchmark. Therefore, we conducted five randomized trainings with non-fixed labeled bags, studying the algorithm's holistic performance across diverse slide representativeness. As suggested by multiple reviewers, we supplemented five runs on Camelyon16 dataset using non-fixed bags, reporting Mean AUC (left panel) and STD (right panel) in Table 1 (bag AUC) and Table 2 (instance AUC). As the shot count increases, standard deviations of all methods tend to stabilize. Our proposed method still achieves the best performance. Mean AUC and STD on the other datasets will be provided in a camera-ready version.
>
> Furthermore, we supplemented five rounds of training using fixed labeled bags, employing the same bag in each run for training. Mean AUC (left panel) and STD (right panel) are presented in Table 3 (bag AUC) and Table 4 (instance AUC). The consistent labeled bags notably decrease variability and our method still achieves the best performance.
>
> **Q2**. Comparison with CoOp using equal learnable parameters
>
> **Response**.
>
> We conducted the experiment on the Camelyon 16 dataset, and the results are presented in Table 6 (mean bag AUC) and Table 7 (mean instance AUC). Experiments marked with the same "*" or "#" have the same learnable parameters.
>
> *a)* We maintained the baseline's parameter size (1. CoOp+Attention pooling) and reduced the tokens in (4. Our approach) and (3. Ablation method) to match the baseline's size. Despite this reduction, our method still outperformed others marked with "*" in Table 6 and Table 7.
>
> *b)* Comparing (5. CoOp+Prompt guided pooling) and (6. Our approach) with equal parameter size (#), our method again achieved superior performance in Table 6 and Table 7.
>
> *c)* Although reducing the parameter size led to some performance degradation, our approach consistently achieved the best performance.
>
> **Q3**. Ablation study concerning the auxiliary loss.
>
> **Response**.
>
> The auxiliary loss aims to separate instance prototypes learned by each instance prompt, ensuring distinct phenotypes representing WSIs. Crucial instance prototypes for slide classification stand out during aggregation. New ablation and sensitivity tests on the Camelyon16 dataset (Table 5) demonstrate that our method is not highly sensitive to the loss weight, but its addition significantly improves performance compared to not using it.

---

### Official Review · Reviewer_VPd6 · 2023-07-21

**Soundness:** 2 fair
**Presentation:** 3 good
**Contribution:** 2 fair
**Rating:** 5
**Confidence:** 5

**Summary:**

This paper introduces an innovative problem setting known as Few-shot Weakly Supervised WSI Classification (FSWC), wherein the availability of labeled WSIs is severely limited. In order to address this challenge, the authors propose a novel Two-level Prompt Learning MIL framework, named TOP, which leverages the power of the VL model and GPT-4 to enhance the learning process of the model.

**Strengths:**

1.	This paper demonstrates exceptional writing quality, distinguished by its clear and well-organized structure. The authors adeptly utilize intuitive figures and well-crafted paragraphs to effectively summarize their methods and contributions. Furthermore, the inclusion of extensive experiments and detailed ablation studies serves as a robust validation for their proposed method.
2.	The proposed instance prompt guided pooling is a captivating and fitting approach for the vision language model task. It exhibits an intriguing method that holds potential for improving the performance of the model in this context.


**Weaknesses:**

1.	The problem addressed in this paper is not entirely novel within the field of WSIs, as previous studies like MI-Zero[1] have also recognized the challenge of limited labeled WSIs.
2.	The utilization of pathology language prior knowledge derived from GPT-4 is indeed an interesting aspect of this paper. However, it is essential to address the concerns and questions raised regarding the validity and importance of this knowledge in the task at hand. To assess the correctness and relevance of the knowledge obtained from GPT-4, the paper should ideally outline an evaluation method or standards based on the output of GPT-4. This would help establish the reliability and usefulness of the information derived from the model. Additionally, the authors should clarify their rationale for choosing GPT-4's knowledge over manually designed and professionally validated visual descriptions. It would be beneficial to discuss how they compared the performance and efficacy of the two approaches and why GPT-4-based knowledge was deemed more appropriate. Regarding the generation of questions for GPT-4, it is crucial to provide insights into the process of selecting and designing these questions. The paper should clarify whether there were any specific criteria or standards used to ensure the validity and reasonableness of the questions. Addressing potential biases introduced by the question selection process is also important. Considering that GPT-4 may undergo updates over time, it is essential to acknowledge the potential impact on the answers generated by the model. The authors should discuss the implications of model updates on the overall performance and reproducibility of their proposed framework. In summary, addressing these questions and providing more detailed explanations would enhance the clarity and credibility of the paper, particularly concerning the validity and reliability of the pathology language prior knowledge derived from GPT-4.
3.	The technical novelty of the paper is acknowledged to be limited. Although the paper introduces the instance prompt guided pooling method, which is derived from prototype learning and presents some novelty, it may not be considered sufficient to significantly differentiate it from existing approaches. Furthermore, it appears that the use of a Vision Language (VL) model to extract features and the utilization of learnable prompts from CoOp are commonly employed techniques within the field[1]. These aspects may not contribute significantly to the novelty of the proposed framework. Considering these limitations, it would be valuable for the authors to address the relative novelty of their contributions more explicitly. It could be helpful to discuss how their approach builds upon or improves existing methods, and to identify the specific aspects that differentiate their work from prior research in the field.
4.	I have a concern regarding the comparison and reporting of results in the paper. Randomly training the network multiple times with different labeled bags and reporting the highest performance of each method may introduce some ambiguity and make the results appear misleading or tricky. To ensure a more robust and fair comparison, it would be preferable to adopt a standardized evaluation methodology. This could involve training and testing the network multiple times using consistent sets of labeled bags for each method, and then reporting the average performance across these trials. This approach would provide a more reliable representation of the methods' benefits and allow for a more accurate comparison between them. By employing a standardized evaluation methodology, the authors would enhance the credibility and trustworthiness of their reported results, addressing concerns about potential biases or inconsistencies introduced by the current approach.


**Questions:**

It would be greatly appreciated if the author could consider making the code, trained models, and specific train-validation-test data splits publicly available for the associated public datasets used in the different methods. This step is crucial for ensuring the reproducibility of the results. Access to these resources would significantly facilitate the verification and replication of the finding. Other questions please refer to the weekness part.

**Limitations:**

Please refer to the weekness part.

---

> ### Author Rebuttal · Authors · 2023-08-09
>
> **Q1 & Q3**. Novelty compared with MI-Zero.
>
> **Response**.
>
> Our approach differs significantly from MI-Zero [1] in the following ways:
>
> *a)* MI-Zero focuses on training a large model like CLIP for instance-level or slide-level zero-shot transfer and classification, while we leverage CLIP as a backbone for instance and slide-level few-shot learning from a prompt-learning perspective. Our concepts and experimental approaches are distinct.
>
> *b)* As highlighted by reviewer 5SeZ in "Strengths", our work explores both instance and bag-level prompts, while MI-Zero only explores "instance-level" prompts and uses simple top-k pooling solely for bag-level classification.
>
> Our work presents notable innovations:
>
> *a) Conceptually*, we address the novel Few-Shot Weakly Supervised WSI Classification (FSWC) problem and pioneer bag and instance-level prompt learning with large vision-language model (CLIP) and large language model (GPT-4) in WSI classification.
>
> *b) Technically*, we propose a Two-level Prompt Learning MIL framework. At the instance level, we use pathology language prior knowledge from GPT-4 to guide feature aggregation into bag features. At the bag-level, we introduce bag-level pathology categories and visual pathology descriptions as prior knowledge for few-shot learning supervised by bag labels. These ideas are fundamentally different from previous studies.
>
> **Q2**. Validity, rationale, motivation and potential model updates' impact for using GPT-4’s knowledge.
>
> **Response**.
>
> *a) Validity and rationale*: We rigorously reviewed pathology knowledge descriptions from GPT-4 with three senior pathologists and found them accurate and detailed. Literature [2] supports GPT-4's reliability in producing medical domain knowledge due to its vast medical expertise in training data.
>
> *b) Motivation and importance*: Leveraging GPT-4's knowledge as templates enhances efficiency versus manual design. This approach aligns with the few-shot learning goal, easing pathologist annotation. Manual templates might not cover all aspects; specialized doctors' templates could be needed for varied cancer types/tasks. Additionally, different doctors' descriptions vary, lacking a standardized manual description. By leveraging GPT-4's versatility, our aim is to attain knowledge descriptions for multiple cancer types and tasks while avoiding manual domain biases.
>
> *c) Impact of model updates*: GPT-4's language descriptions contributed to training pathology models in our research. We will publicly share all used descriptions, codes, and models. This disclosure ensures reproducibility in reported tasks without the need of invoking GPT-4 for inference or new training. GPT-4’s upgrades won't influence current outcomes. We'll explore if GPT upgrades generate new descriptions and their effect on results.
>
> **Q4**. Consistent labeled bag sets for each method.
>
> **Response**.
>
> Firstly, "training five times with non-fixed labeled bags" in the original text means random selection of different labeled bags in each run based on shot count to compose the training set. The test set is pre-divided and remains unseen during training. All comparative methods use the same training and test sets in each run, ensuring fair comparisons.
>
> In the few-shot learning setup, limited labeled bags for training can cause performance disparities due to varying bag representativeness (like selecting slides with larger or smaller tumor areas). This leads to high standard deviation (STD) across runs with different labeled bags for training. This work is one of the first few-shot WSI learning studies, and there is not a relevant benchmark. Therefore, we conducted five randomized trainings with non-fixed labeled bags, studying the algorithm's holistic performance across diverse slide representativeness. As suggested by multiple reviewers, we supplemented five runs on Camelyon16 dataset using non-fixed bags, reporting Mean AUC (left panel) and STD (right panel) in Table 1 (bag AUC) and Table 2 (instance AUC). As the shot count increases, standard deviations of all methods tend to stabilize. Our proposed method still achieves the best performance. Mean AUC and STD on the other datasets will be provided in a camera-ready version.
>
> Furthermore, we supplemented five rounds of training using fixed labeled bags, employing the same bag in each run for training. Mean AUC (left panel) and STD (right panel) are presented in Table 3 (bag AUC) and Table 4 (instance AUC). The consistent labeled bags notably decrease variability and our method still achieves the best performance.
>
> [1] Lu et al. Visual Language Pretrained Multiple Instance Zero-Shot Transfer for Histopathology Images, CVPR. (2023).
>
> [2] Nori et al. Capabilities of gpt-4 on medical challenge problems. arXiv:2303.13375 (2023).
>
> **Q5**. Code, data and model availability.
>
> **Response**.
>
> We will publicly share all used descriptions, codes, and models.

---

> > ### Comment · Reviewer_VPd6 · 2023-08-12
> >
> > Thanks the authors for supplementing five rounds of training using fixed labelled bags, employing the same bag in each run for training. Mean AUC (left panel) and STD (right panel) are presented in Table 3 (bag AUC) and Table 4 (instance AUC). The answer to my "Validity, rationale, motivation and potential model updates' impact for using GPT-4’s knowledge." also makes sense, so I would like to make the new recommendation of the paper as "acceptance beyond marginal".

---

> > > ### Author Response · Authors · 2023-08-12
> > > **Gratitude for Your Review**
> > >
> > > Thank you for reviewing our paper. Your comments are greatly appreciated and will undoubtedly enhance the quality of our research. We look forward to revising our paper and submitting an improved version.

---

> > > > ### Comment · Reviewer_5SeZ · 2023-08-12
> > > > **Comparison to MI-Zero**
> > > >
> > > > From reading the comments of Reviewer VPd6 and the reply, while it is true that "MI-Zero focuses on training a large model like CLIP for instance-level or slide-level zero-shot transfer and classification", why were quantitative comparisons to MI-Zero not made, as MI-Zero is the only comparable zero-shot slide classification method to this work. I partially agree with the innovations proposed in your reply, but these innovations need to be carefully ablated and studied w.r.t. to currently-used baselines.

---

### Author Rebuttal · Authors · 2023-08-09

We sincerely thank all reviewers for your valuable comments. We first reply to questions raised by multiple reviewers and then other questions from every reviewer. We have uploaded all seven tables into a PDF file, and we sincerely appreciate your downloading and reviewing them.

**Q1**. Average AUC for 5 runs instead of max AUC. (Reviewer VPd6, 5SeZ and 7zsM)

**Response**.

Firstly, "training five times with non-fixed labeled bags" in the original text means random selection of different labeled bags in each run based on shot count to compose the training set. The test set is pre-divided and remains unseen during training. All comparative methods use the same training and test sets in each run, ensuring fair comparisons.

In the few-shot learning setup, limited labeled bags for training can cause performance disparities due to varying bag representativeness (like selecting slides with larger or smaller tumor areas). This leads to high standard deviation (STD) across runs with different labeled bags for training. This work is one of the first few-shot WSI learning studies, and there is not a relevant benchmark. Therefore, we conducted five randomized trainings with non-fixed labeled bags, studying the algorithm's holistic performance across diverse slide representativeness. As suggested by multiple reviewers, we supplemented five runs on Camelyon16 dataset using non-fixed bags, reporting Mean AUC (left panel) and STD (right panel) in Table 1 (bag AUC) and Table 2 (instance AUC). As the shot count increases, standard deviations of all methods tend to stabilize. Our proposed method still achieves the best performance. Mean AUC and STD on the other datasets will be provided in a camera-ready version.

Furthermore, we supplemented five rounds of training using fixed labeled bags, employing the same bag in each run for training. Mean AUC (left panel) and STD (right panel) are presented in Table 3 (bag AUC) and Table 4 (instance AUC). The consistent labeled bags notably decrease variability and our method still achieves the best performance.

**Q2**. The motivation, ablation and sensitivity tests of auxiliary loss. (Reviewer 5SeZ and 7zsM)

**Response**.

The auxiliary loss aims to separate instance prototypes learned by each instance prompt, ensuring distinct phenotypes representing WSIs. Crucial instance prototypes for slide classification stand out during aggregation. New ablation and sensitivity tests on the Camelyon16 dataset (Table 5) demonstrate that our method is not highly sensitive to the loss weight, but its addition significantly improves performance compared to not using it.

**Q3**. Validity, rationale, motivation and potential model updates' impact for using GPT-4’s knowledge. (Reviewer VPd6 and r2fX)

**Response**.

*a) Validity and rationale*: We rigorously reviewed pathology knowledge descriptions from GPT-4 with three senior pathologists and found them accurate and detailed. Literature [1] supports GPT-4's reliability in producing medical domain knowledge due to its vast medical expertise in training data.

*b) Motivation and importance*: Leveraging GPT-4's knowledge as templates enhances efficiency versus manual design. This approach aligns with the few-shot learning goal, easing pathologist annotation. Manual templates might not cover all aspects; specialized doctors' templates could be needed for varied cancer types/tasks. Additionally, different doctors' descriptions vary, lacking a standardized manual description. By leveraging GPT-4's versatility, our aim is to attain knowledge descriptions for multiple cancer types and tasks while avoiding manual domain biases.

*c) Impact of model updates*: GPT-4's language descriptions contributed to training pathology models in our research. We'll publicly share all used descriptions, codes, and models. This disclosure ensures reproducibility in reported tasks without the need of invoking GPT-4 for inference or new training. GPT-4’s upgrades won't influence current outcomes. We'll explore if GPT upgrades generate new descriptions and their effect on results.

[1] Nori et al. Capabilities of gpt-4 on medical challenge problems. arXiv:2303.13375 (2023).

---

> ### Author Response · Authors · 2023-08-19
> **Mean AUC and STD on the TCGA and Cervical Cancer datasets**
>
> As promised in our first round of response, we conducted further experiments and report the results of five runs on *the TCGA dataset* and *the Cervical Cancer dataset* using non-fixed bags, reporting Mean (left panel) and STD (right panel) in Table 1 (*the TCGA dataset*) and Table 2 (*the Cervical Cancer dataset*). We also supplemented five rounds of training using fixed labeled bags on *the TCGA dataset* and *the Cervical Cancer dataset*, employing the same bag in each run for training. Mean (left panel) and STD (right panel) are presented in Table 3 (*the TCGA dataset*) and Table 4 (*the Cervical Cancer dataset*). We can draw similar conclusions as those obtained on the Camelyon16 dataset, and our approach demonstrated the best performance across all datasets. All of these results will be included in the camera-ready version.
>
> Table 1: Mean bag AUC using **non-fixed labeled bags** on *the TCGA dataset*
>
> |Method|16-shot|8-shot|4-shot|2-shot|1-shot|16-shot|8-shot|4-shot|2-shot|1-shot|
> |:-------------------------------------------:|:-------:|:-------:|:-------:|:-------:|:-------:|:-------:|:-------:|:-------:|:-------:|:-------:|
> |Linear-Probe (Mean-pooling)| 0.5899| 0.5126| 0.4843| 0.4645| 0.4527| 0.0451| 0.0263| 0.0282| 0.0217| 0.0265|
> |Linear-Probe (Max-pooling)| 0.5951| 0.5245| 0.5037| 0.4731| 0.4615| 0.0512| 0.0563| 0.0647| 0.0365| 0.0174|
> |Linear-Probe   (Attention-pooling)| 0.6868| 0.6223| 0.5954| 0.5569| 0.5524| 0.0279| 0.0516| 0.0699| 0.0492| 0.0596|
> |CoOp (Attention-pooling)| 0.7545| 0.6700| 0.6662| 0.6416| 0.6306| 0.0280| 0.0512| 0.0636| 0.0751| 0.0860|
> | Bag Prompt + Prompt guided pooling (**Ours**) |**0.8039**|**0.7685**|**0.7279**|**0.6984**|**0.6879**|0.0265|0.0465|0.0641|0.0743|0.0852|
>
> Table 2: Mean bag AUC using **non-fixed labeled bags** on *the Cervical Cancer dataset*
>
> |Method|16-shot| 8-shot| 4-shot|2-shot|1-shot|16-shot|8-shot|4-shot|2-shot|1-shot|
> |:-------------------------------------------:|:-------:|:-------:|:-------:|:-------:|:-------:|:-------:|:-------:|:-------:|:-------:|:-------:|
> |Linear-Probe (Mean-pooling)| 0.6514  | 0.6348  | 0.5839  | 0.5740  | 0.5615  | 0.0556  | 0.0696  | 0.0781  | 0.0670  | 0.0634  |
> |Linear-Probe (Max-pooling)| 0.6003  | 0.5889  | 0.5593  | 0.5525  | 0.5343  | 0.0597  | 0.0635  | 0.0660  | 0.0612  | 0.0436  |
> |Linear-Probe   (Attention-pooling)| 0.7052  | 0.6690  | 0.6524  | 0.6496  | 0.5178  | 0.0364  | 0.0624  | 0.0814  | 0.0871  | 0.0564  |
> |CoOp (Attention-pooling)| 0.7041  | 0.6606  | 0.6568  | 0.6417  | 0.6049  | 0.0303  | 0.0658  | 0.0775  | 0.0855  | 0.0931  |
> |Bag Prompt + Prompt guided   pooling (**Ours**) | **0.8080**  | **0.7855**  | **0.7468**  | **0.7123**  | **0.6652**  | 0.0296  | 0.0549  | 0.0733  | 0.0806  | 0.0957  |
>
> Table 3: Mean bag AUC using **fixed labeled bags** on *the TCGA dataset*
>
> |                    Method                   | 16-shot |  8-shot |  4-shot |  2-shot |  1-shot | 16-shot |  8-shot |  4-shot |  2-shot |  1-shot |
> |:-------------------------------------------:|:-------:|:-------:|:-------:|:-------:|:-------:|:-------:|:-------:|:-------:|:-------:|:-------:|
> |         Linear-Probe (Mean-pooling)         | 0.6022  | 0.5418  | 0.4934  | 0.4908  | 0.4646  | 0.0067  | 0.0072  | 0.0084  | 0.0092  | 0.0126  |
> |          Linear-Probe (Max-pooling)         | 0.6227  | 0.5547  | 0.5155  | 0.4985  | 0.4876  | 0.0065  | 0.0072  | 0.0089  | 0.0090  | 0.0137  |
> |      Linear-Probe   (Attention-pooling)     | 0.7178  | 0.6539  | 0.6248  | 0.5832  | 0.5713  | 0.0067  | 0.0071  | 0.0072  | 0.0082  | 0.0107  |
> |           CoOp (Attention-pooling)          | 0.7840  | 0.6824  | 0.6811  | 0.6772  | 0.6801  | 0.0064  | 0.0065  | 0.0084  | 0.0095  | 0.0105  |
> | Bag Prompt + Prompt guided   pooling (**Ours**) | **0.8235**  | **0.8059**  | **0.7531**  | **0.7245**  | **0.7123**  | 0.0059  | 0.0062  | 0.0083  | 0.0092  | 0.0103  |
>
> Table 4: Mean bag AUC using **fixed labeled bags** on *the Cervical Cancer dataset*
>
> |Method| 16-shot |  8-shot |  4-shot |  2-shot |  1-shot | 16-shot |  8-shot |  4-shot |  2-shot |  1-shot |
> |:-------------------------------------------:|:-------:|:-------:|:-------:|:-------:|:-------:|:-------:|:-------:|:-------:|:-------:|:-------:|
> |         Linear-Probe (Mean-pooling)         | 0.6756  | 0.6684  | 0.6593  | 0.6246  | 0.6011  | 0.0098  | 0.0105  | 0.0104  | 0.0113  | 0.0147  |
> |          Linear-Probe (Max-pooling)         | 0.6322  | 0.6249  | 0.6038  | 0.5884  | 0.5869  | 0.0103  | 0.1120  | 0.0106  | 0.0114  | 0.0124  |
> |      Linear-Probe   (Attention-pooling)     | 0.7345  | 0.7282  | 0.7155  | 0.6873  | 0.6137  | 0.0095  | 0.0105  | 0.0108  | 0.0169  | 0.0155  |
> |           CoOp (Attention-pooling)          | 0.7565  | 0.7349  | 0.7271  | 0.6927  | 0.6484  | 0.0091  | 0.0095  | 0.0097  | 0.0115  | 0.0117  |
> | Bag Prompt + Prompt guided   pooling (**Ours**) | **0.8189**  | **0.8007**  | **0.7869**  | **0.7618**  | **0.7052**  | 0.0088  | 0.0089  | 0.0088  | 0.0104  | 0.0101  |

---

### Decision · Program_Chairs · 2023-09-21

**Decision:**

Accept (poster)

**Comment:**

While some concerns about the evaluation of the method remain, the ideas on how to generate guidance in the few shot bag-level scenario are novel and will potentially have an impact on the field. The authors responses and comparisons appear satisfactory to the AC.